# Degraded Polygons Raise Fundamental Questions of Neural Network Perception

**Leonard Tang**
Department of Mathematics
Harvard University
Cambridge, MA 02138
leonardtang@college.harvard.edu

**Dan Ley**
School of Engineering and Applied Sciences
Harvard University
Cambridge, MA 02134
dley@g.harvard.edu

## Abstract

It is well-known that modern computer vision systems often exhibit behaviors misaligned with those of humans: from adversarial attacks to image corruptions, deep learning vision models suffer in a variety of settings that humans capably handle. In light of these phenomena, here we introduce another, orthogonal perspective studying the human-machine vision gap. We revisit the task of recovering images under degradation, first introduced over 30 years ago in the Recognition-by-Components theory of human vision. Specifically, we study the performance and behavior of neural networks on the seemingly simple task of classifying regular polygons at varying orders of degradation along their perimeters. To this end, we implement the Automated Shape Recoverability Test[1] for rapidly generating large-scale datasets of perimeter-degraded regular polygons, modernizing the historically manual creation of image recoverability experiments. We then investigate the capacity of neural networks to recognize and recover such degraded shapes when initialized with different priors. Ultimately, we find that neural networks' behavior on this simple task conflicts with human behavior, raising a fundamental question of the robustness and learning capabilities of modern computer vision models.

## 1 Introduction

Since the advent of adversarial attacks (Goodfellow et al., 2015), researchers have grown increasingly wary of machine learning models' susceptibility to learning irrelevant patterns (Yuan et al., 2019). Oftentimes, neural networks rely on spurious features that humans know to avoid (Khani and Liang, 2021). A poignant example of such unorthodox behavior comes in the form of machine vision's over-dependency on object textures rather than object shapes (Geirhos et al., 2019). This often leads to dangerous consequences in practice (Hendrycks et al., 2021a). Similarly, the fragility of vision models in response to minor image transformations such as shifts or rotations (Azulay and Weiss, 2019), raises concerns over how well these models truly learn, especially when considering these small geometric transformations are commonplace in natural vision scenarios. The specifics of when or how vision models might be expected to generalize well remains a mystery (Zhang et al., 2017).

Beyond being unreliable in real-world settings, current vision models are decidedly unhuman in nature. They tend to learn undesirable features that are fundamentally misaligned with how humans perceive the world. Given the unexpected nature of such models, we study if models are also capable of correctly identifying recoverable images, a concept first presented in the Recognition-by-Components theory of human vision (Biederman, 1987). Similarly, we investigate whether or not these models' performance is marred by non-recoverable images.

---

[1]https://github.com/leonardtang/Degraded-Polygons-RBC

37th Conference on Neural Information Processing Systems (NeurIPS 2023) Track on Datasets and Benchmarks.

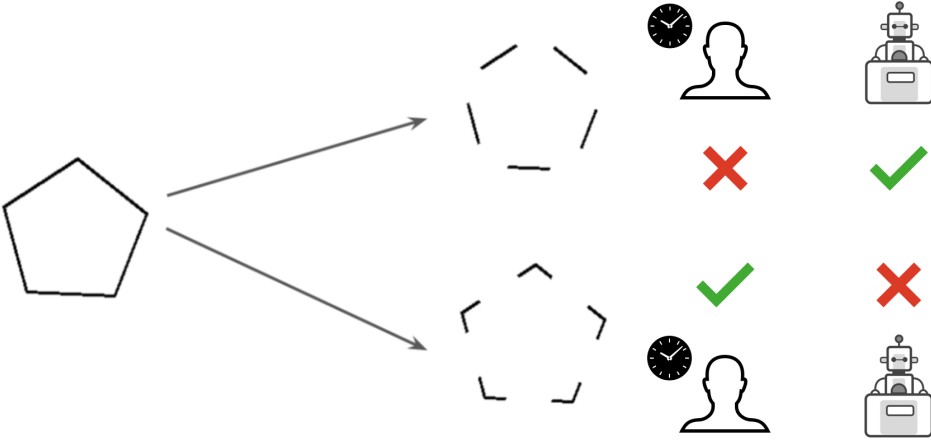

Figure 1: Specific instance of the Automated Shape Recoverablility Test generation pipeline for an example pentagon with 50% degradation proportion. Whole shapes are generated and subsequently edited with corner degradation (top), and edge degradation (bottom). Our experiments indicate that, unlike time-constrained humans performing sketch recovery (Biederman, 1987), neural networks rely heavily on edges rather than corners to recover degraded shapes.

To make progress in understanding the behavior of computer vision models, we introduce the Automated Shape Recoverablility Test pipeline for evaluating vision models across a spectrum of image degradation in regular polygons. The intrinsic difficulty in classifying a generated image is directly controlled by the proportion of the image we delete, subject to the constraint of where the image deletion is allowed to occur. We operate on the domain of black-and-white sketches, since they most closely resemble the distribution of images presented in Biederman (1987), which consist of simple sketches of common objects. Moreover, our ultimate results on this simple task setting suggest a fundamental misalignment in the way humans and machine approach image classification.

Using our pipeline, we produce 1,260,000 sketches of regular polygons evenly distributed across 7 shape categories, 9 levels of image degradation, and 2 forms of degradation (corner and edge degradation). Though seemingly simple, these images measure model performance against a canonical human vision task, yielding surprising discrepancies. We release the editing pipeline and final dataset of 1,260,000 images in the hopes of encouraging further research in this direction.

In image classification experiments on a subset of the data, we observe that common vision architectures poorly recover (i.e. correctly classify) heavily edge-degraded and corner-degraded shapes, both of which humans are capable of recognizing. Surprisingly, neural networks also rely primarily on edges rather than corners for shape recovery – the exact opposite of human behavior. Moreover, our results also indicate that models pretrained entirely on non-accidental properties generated by Iterated Function Systems (Barnsley and Vince, 2010) display much stronger performance patterns on the same corner-removed class. Overall, our contributions are summarized as follows:

- We introduce the Automated Shape Recoverability Test, a pipeline for generating datasets with parameterized degradation of regular polygons. We publicly release the pipeline and the accompanying dataset of 1,260,000 images, which span across seven categories and include varying degrees and forms of image degradation.

- Through a comprehensive analysis of various neural network architectures on the task of shape recovery across varying levels of image degradation, we demonstrate a striking discrepancy in how machine learning models and humans perceive images. Unlike humans, neural networks consistently rely more on edges than corners for image recovery, pointing to a fundamental difference in image processing between machines and humans.

- Our exploration of pretraining dataset choice reveals that models pretrained on fractals, unlike those pretrained on ImageNet, retain greater accuracy on edge-degraded shapes, while continuing to perform poorly on corner-degraded shapes, further misaligning human and machine vision. Through Grad-CAM visualizations, we reveal differences in how ImageNet and fractal pretrained models learn features and process degraded shapes.

## 2    Related Work

### 2.1    Sketch Classification

Though sketch classification is not as common a task as natural image classification, much existing research attempts to tackle the problem. The TUBerlin Sketch Benchmark was the first large-scale sketch classification dataset for machine learning, consisting of 20,000 unique sketches distributed over 250 object categories that exhaustively cover most objects that are commonly encountered in everyday life Eitz et al. (2012). 1,350 unique Amazon Mechanical Turk (MTurk) workers were recruited to sketch these images, with each worker only being able to sketch a limited number of sketches per category so as to preserve diversity within each sketch class. Eitz et al. (2012) also develop a bag-of-features sketch representation alongside multi-class support vector machines trained on the dataset, which classifies unseen sketches with 56% accuracy.

Since the introduction of this dataset, significant work has been done to build more accurate models of sketch classification using deep-learning architectures, particularly Convolutional Neural Networks and Recurrent Neural Networks, as well as solving challenges including partial sketch classification and sketch progression incorporation (Tran, 2017; Seddati et al., 2015, 2016; Yang and Hospedales, 2015; Ha and Eck, 2018). As a result of these efforts, the competitive accuracy on the TUBerlin Sketch Benchmark is now 77.69%, a significant improvement over the original paper.

### 2.2    Image Recovery from a Cognitive Science Perspective

**Recognition-by-Components**    The task of *image recovery* was first introduced in the landmark Recognition-by-Components theory from cognitive science, which explains how humans recognize and categorize objects based on their basic geometric structures, called geons, and their spatial relationships (Biederman, 1987; Cooper and Biederman, 1993). A critical component of Recognition-by-Components theory states that object recognition of two-dimensional black-and-white sketches is impaired when feature-relation information – the information about the relationships between geons – is degraded. In the limit of feature-relation information degradation, it is ambiguous what the original object was supposed to be; contextual inference of the original object is no longer possible. We define such a degraded image to be *non-recoverable*. Otherwise, a degraded image that can still be recognized is *recoverable*.

Recognition-by-Components also posits that parsing of an object into components is performed at vertex regions of sharp concavity, with multiple curves terminating at a common point. To verify this claim, Biederman (1987) tested humans' ability to classify sketches of objects in a timed setting after object contours had undergone heavy degradation. Expectedly, heavier degradation – greater proportions of the sketch being deleted – led to lower classification accuracy. Additionally, corroborating the original claim, degradation centered at object corners induced much lower accuracy in subjects compared to degradation along curves between corners (Biederman, 1987; Guez et al., 1994).

While our proposed dataset and task do not exactly match the setting of Biederman's (1987) experiments in the sense that we focus on a distinct set of sketch shapes and do not consider time-constrained recognition, our results indicate that this task is sufficiently challenging for modern deep learning models. Moreover, the simple structure of our data, being merely regular polygons, indicates a fundamental misalignment of deep learning vision models with respect to human vision. We view our dataset as a minimum viable task that highlights this misalignment: the fact that deep learning models exhibit pathologies even on this simple shape classification task suggests that neural network perception may be even less robust than we know. To contextualize our findings, we compare Biederman's (1987) human subject performance to those of common vision architectures in §4.1.

**Gollin Figures Test**    Closely related to image recovery is the Gollin figures test for assessing human visual perception. Subjects are shown variations of a common object in quick secession, with five consecutive incomplete line drawings of a picture, from least to most complete, displayed to each subject (Gollin, 1960). The subject needs to mentally complete the underlying drawing in order to identify the original object drawn, similar to testing for Gestalt closure phenomena (Rock and Palmer, 1990; Kim et al., 2021). Notably, unlike in Recognition-by-Components theory, the Gollin figures test makes no distinction between image degradations that yield recoverable images and degradations that yield non-recoverable images.

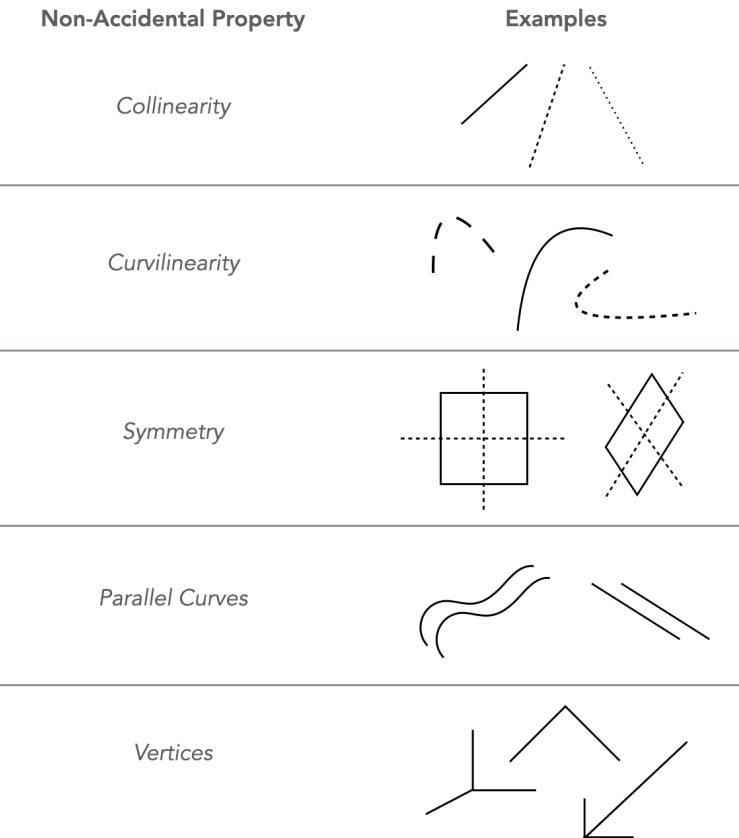

| Non-Accidental Property | Examples |
|---|---|

Collinearity

Curvilinearity

Symmetry

Parallel Curves

Vertices

Figure 2: The five classes of nonaccidental properties (NAPs) for object recognition in the visual cortex are 1) *collinearity*, the presence of straight lines; 2) *curvilinearity*, the presence of smoothly curved elements; 3) *symmetry* across arbitrary axes; 4) *parallel curves*; and 5) *vertices*, junctions of two or more contours (Lowe, 1985). Critically, cognitive scientists suggest that NAPs form a perceptual basis for the set of components that enable object recognition.

Historically, the Gollin figures test has been automated in order to provide more fine-grained control over the proportion of an image that is degraded. That is, rather than having five discrete variants of degradation as in the original test, software was introduced to construct infinitely many variants of degradation on a continuous spectrum (Foreman and Hemmings, 1987). Motivated by this precedent, we automate the generation of recoverable and non-recoverable image degradations at scale.

**Non-Accidental Properties**   Non-accidental properties (NAPs) are image properties that are invariant over orientation and depth (Amir et al., 2012). The human visual system processes NAPs in a two-dimensional drawing as feasibly occurring in three dimensions (Witkin and Tenenbaum, 1983). For example, if there is a straight line, a manifesation of *collinearity*, in a two-dimensional drawing, the visual system infers that the edge producing that line in the three-dimensional world is also straight. In other words, NAPs are dimension-invariant. Another perspective for understanding NAPs is the *non-accidentalness principle*, which states that spatiotemporal coherence and regularity is so unlikely to arise by the random interaction of independent components, that such structure, when observed, almost certainly denotes an underlying unified process (Blusseau et al., 2016). NAPs precisely capture these unlikely coherences, providing useful cues for human object recognition. Figure 2 displays the five canonical NAPs and examples of each property (Lowe, 1985).

NAPs and image recoverability are intimately related notions. Specifically, NAP location and type directly parameterizes regions of non-recoverable image degradations. If NAPs are removed from any image region, it becomes more difficult to recover the object component that the deleted NAP belongs too. Under certain NAP removals, it becomes more difficult to recover the relationship between object components, and thus the overall image. For instance, Biederman (1987) demonstrated that deletion of vertices adversely affected object recognition more than deletion of midsegments. Inspired

by this result, here we focus on producing degradation of regions surrounding vertices as well as midsegments in order to generate non-recoverable and recoverable images, respectively.

**Fractals as Partial Non-Accidental Properties**    While there are not any known ways of specifying the generation of NAPs directly, we draw inspiration from Barnsley and Vince (2010) to automatically generate *fractals*. Due to their intrinsic collinearity, curvilinearity, symmetry, parallel nature, multitude of junctions, and occurrence in natural objects and scenes, fractals may be regarded as a proxy form of NAPs. To that end, we investigate the performance of models pretrained via fractal-guided Formula-driven Supervised Learning (FSDL) on our benchmark. In FSDL, fractals are generated by mathematical formulae, then binned into categories based on the range of formula parameters. The learning task is thus to classify fractals within the correct parameter range (Kataoka et al., 2020).

Recent results in the deep learning literature also motivate the use of fractal pretraining. In particular, Hendrycks et al. (2021b) showed that a simple data augmentation technique mixing fractals with natural images comprehensively improves model robustness and safety metrics. Contemporaneously, Kataoka et al. (2020) also demonstrate that pretraining neural networks entirely on synthetically generated fractals achieves the same level of performance on downstream tasks as pretraining on natural images. For our experiments, we use models pretrained on synthetic fractals with 10,000 fractal classes, which we refer to as FractalDB models.

## 3    Automating Shape Recoverability Experiments

We implement the Automated Shape Recoverablility Test, an automatic image editing pipeline for creating degraded regular polygons at varying severities. Figure 1 displays an example of our generation pipeline. In total, our pipeline requires less than 30 seconds of compute time to serially generate 6,000 diverse images across 7 shape categories – triangles, squares, pentagons, hexagons, heptagons, and octagons – with 1,000 images each[2]. Moreover, this procedure can be trivially scaled up to any arbitrary number of polygon classes and images per class. Our approach broadly consists of generating regular polygons, then degrading their perimeters to produce recoverable (edge degradation) and non-recoverable (corner degradation) images.

**Regular Polygon Generation**    Our shape generation pipeline begins by constructing regular polygons. We generate polygons by first implicitly defining a circle, then placing an appropriate number of points, matching the desired number of sides, on its circumference. For example, to draw a pentagon, we place five points equally spaced on the circumference of the circle, then use line segments to connect them. To promote image diversity, the center of the circle $c$ is chosen randomly within a 224 $\times$ 224 grid, subject to a minimum acceptable radius size $r_{min}$. Subsequently, the circle's radius $r$ is chosen uniformly at random between the minimum accepted radius size and the maximum radius size allowed by the sampled circle center. Furthermore, shapes are rotated uniformly at random by an angle $\theta$ between $0°$ and $360°$. The border of each polygon is fixed to be two pixels thick. All generated shapes have black borders overlaid on top of a white background.

**Producing Degraded Shapes**    In order to produce recoverable and non-recoverable versions of our shapes, we parametrically and evenly degrade the perimeter of each polygon. To do so, we first specify the proportion $p_d$ of the shape's perimeter to degrade. Naturally, a larger degradation proportion yields more occluded shapes that are more difficult to recognize, and a smaller degradation proportion yields shapes closer to the original polygon, which are more easily recognized. To produce non-recoverable shapes, we degrade perimeter regions surrounding each corner, and to produce recoverable shapes, we degrade perimeter regions from the midsegment of each edge.

**Corner Degradation**    To degrade corners from an initial regular polygon generation, we overlay white circles at each corner, effectively erasing the perimeter of the corresponding local neighborhood around the corner. Given a desired global degradation proportion $p_d$, each individual white circle is then defined with the following radius, where $N_s$ is the number of sides of the regular polygon and $P$ is the total perimeter:

$$r = \frac{p_d}{2 \cdot N_s} \cdot P$$

---

[2]Benchmarked on an 8-core Apple M1 Pro chip.

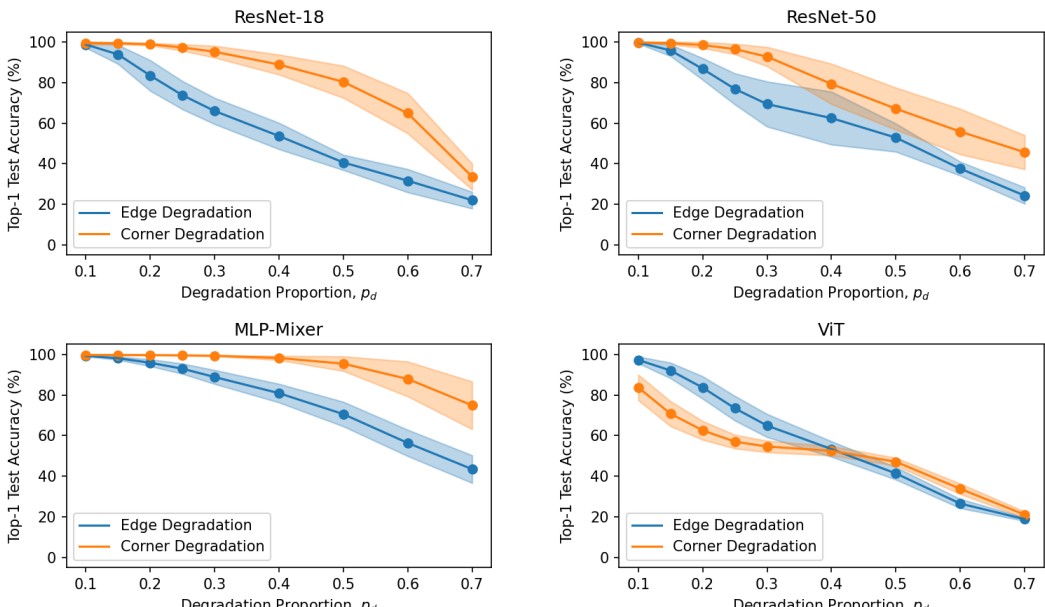

Figure 3: Top-1 test accuracy (%) within $\pm 1$ SD (across 10 training trials) of ImageNet-pretrained and whole-polygon finetuned models on the shape recovery task. Accuracy decreases as degradation proportion, $p_d$, increases. Moreover, ResNet-15, ResNet-50, and MLP-Mixer all exhibit worse performance on edge-degraded compared to corner-degraded shapes, the opposite of human behavior.

Under this construction, the circle at each corner will erase $2r = (P \cdot p_d)/N_s$ pixels from the shape's perimeter, and in aggregate $N_s$ circles will erase $P \cdot p_d$ pixels, thus precisely degrading a $(P \cdot p_d)/P = p_d$ proportion of the original image, as desired.

**Edge Degrading** We adopt a similar procedure for edge degradation. However, instead of overlaying circles at shape corners, we overlay circles at midpoints between corners. Critically, defining $r$ exactly as above, observe that no circle drawn at a midpoint can erase a corner so long as $p_d < 1$, which is true for all of our experiments and in all meaningful degradation cases. In the limiting cases, $p_r = 1$ erases the entire perimeter of the shape, and $p_r = 0$ retains the shape in full. Notably, this procedure performs a single removal at the middle of each edge, *not* a dashed-line edit.

## 4 Experiments

Our experiments evaluate neural networks' ability to classify degraded shapes at 10%, 15%, 20%, 25%, 30%, 40%, 50%, 60%, and 70% perimeter degradation proportions. For our architectures, we benchmark ResNet-18, ResNet-50 (He et al., 2016), MLP-Mixer (Tolstikhin et al., 2021), and ViT (Dosovitskiy et al., 2021). We consider the ResNet family due to its universality in vision experiments and relative simplicity. We also test MLP-Mixer and ViT due to their qualitatively different learning procedure and behavior. MLP-Mixer uses simple multi-layer perceptrons (MLPs) to mix features locally and globally, contrasting the explicit weight sharing seen in convolutional and transformer models. ViT adapts the transformer architecture from natural language processing to handle images, learning global dependencies by treating image patches as a sequence.

We initialize our models using pretrained weights derived from ImageNet (Deng et al., 2009) and FractalDB-10k, a database without any natural images (Kataoka et al., 2020). All models are then finetuned on *whole*, non-degraded regular polygons according to a 60%/20%/20% training/validation/test split. Subsequently, they are directly tested on corner-degraded and edge-degraded polygons at vary-

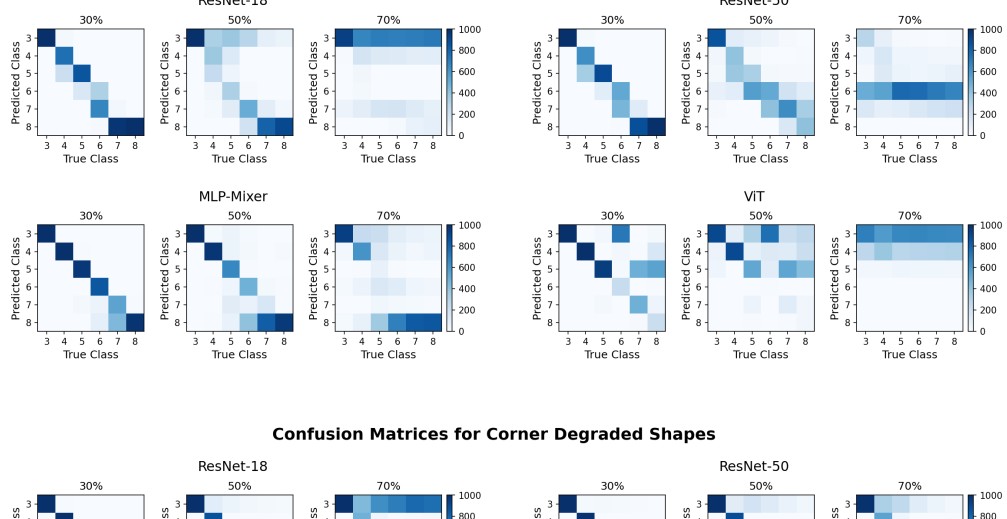

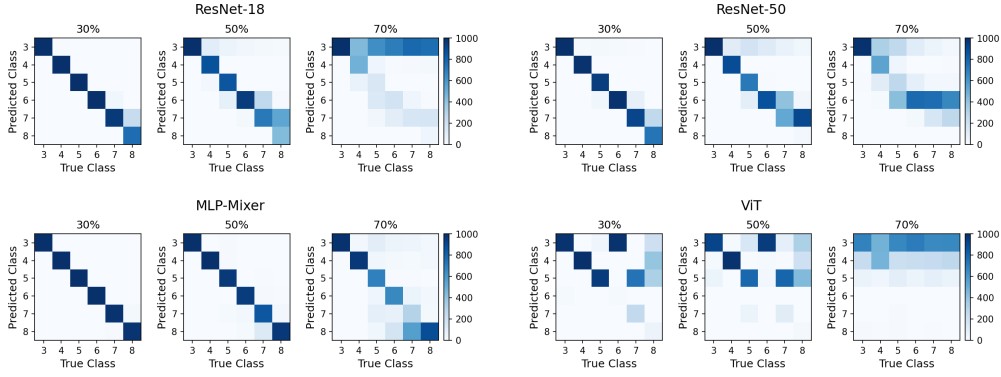

Figure 4: Confusion matrices at 30%, 50%, and 70% removal proportions for corner-degraded shapes (top) and edge-degraded shapes (bottom) using ImageNet-pretrained ResNet-18, ResNet-50, MLP-Mixer, and ViT. As removal proportion increases, models default to predicting a single class.

ing degradation strengths. While our models obtain high test accuracy on whole shape classification, we are chiefly interested in their performance on degraded shapes.

**Training Setup** We finetune ResNet-18 and ResNet-50 on whole polgyons for 20 epochs using SGD with a learning rate of 0.01, a momentum term of 0.9, no weight decay, and a batch size of 64. The learning rate follows a Reduce-Learning-Rate-on-Plateau scheduler with a patience of 3 epochs and a learning rate reduction factor of 0.1. For ViT, we use SGD with a learning rate of 0.001, weight decay of 0.001, momentum of 0.9, and a batch size of 32. Across all experiments, standard data augmentation and preprocessing techniques are used, namely random cropping, random rotations, random horizontal flipping, and normalization over the whole-polygon dataset.

At every epoch, we compute top-1 and top-5 validation accuracy on the whole-polygon dataset, and the weights of the network are saved. After training, the set of weights with the highest top-1 validation accuracy is used to compute test accuracy on whole polygons. We then evaluate this best model on corner-degraded and edge-degraded shapes. Critically, our models achieve 100% validation accuracy within 10 epochs, and also consistently achieve greater than 99.7% accuracy on randomly held out test sets of whole shapes. More interestingly, they perform much worse on degraded shapes.

**Comparison Across Architectures** The top-1 test accuracies on corner-degraded and edge-degraded shapes for ImageNet-pretrained models are shown in Figure 3. Unsurprisingly, as degradation proportion increases, all architectures perform worse, both in the corner-degraded and edge-degraded settings. While ViT performs similarly on all degraded shapes at each degradation proportion, ResNet-18, ResNet-50, and MLP-Mixer consistently perform worse on edge-degraded shapes

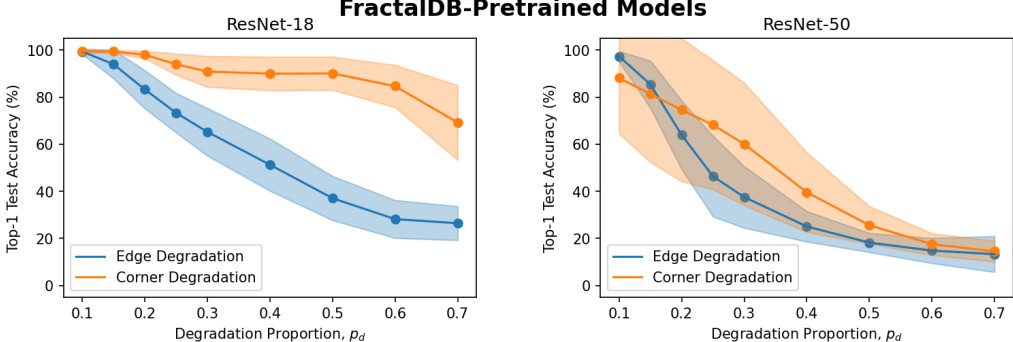

Figure 5: Top-1 test accuracy (%) within $\pm 1$ SD (across 10 training trials) of FractalDB-pretrained and whole-polygon finetuned models on the shape recovery task. Again, accuracy decreases across the board as degradation proportion, $p_d$, increases. Compared to their ImageNet-pretrained counterparts, however, ResNet-18 and ResNet-50 both retain better performance on corner-degraded shapes. We also note the discrepancy compared to edge-degraded shapes, the opposite of human behavior.

versus corner-degraded shapes. Notably, this is the opposite of how time-constrained humans perform on Biederman's (1987) degraded sketch recognition task.

Figure 4 also presents confusion matrices for ImageNet-pretrained ResNet-18, ResNet-50, MLP-Mixer, and ViT at 30%, 50%, and 70% degradation. As degradation proportion increases, we note that models tend to default to predicting a single class, for example to defaulting to triangles for ResNet-18 and hexagons for Resnet-50 at 70% degradation.

## 4.1 Results

**Human Comparison** To contextualize the behavior of neural networks on our shape recovery task, we compare the performance of MLP-Mixer against human subjects from Biederman's (1987) original image recovery experiments. There, image degradation at corners and edges was performed for 18 objects at degradation proportions of 25%, 45%, and 65%. Degraded objects were then exposed to subjects for 100msec, 200msec, and 750msec. Figure 6 compares MLP-Mixer performance against these human subjects in the 100msec setting. Specifically, we compute the percentage difference in accuracy on edge-degraded shapes relative to corner-degraded shapes for both humans and MLP-Mixer. As degradation proportion increases, human subjects show greater accuracy preservation on edge-degraded shapes compared to corner-degraded shapes. Conversely, MLP-Mixer retains increasingly higher accuracy on corner-degraded shapes relative to edge-degraded shapes.

**Dataset Priors** We also investigate the effect of pre-training dataset choice on our models' performance. Besides ImageNet, we also pretrain ResNet-18 and ResNet-50 on FractalDB before finetuning them on our whole-polygon dataset. Figure 5 displays the performance of these models on the shape recovery task. Tables 1 and 2 show the analogous per class

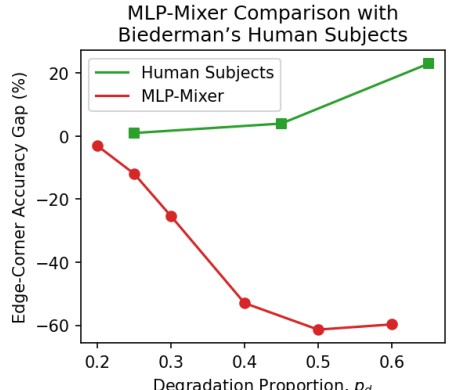

Figure 6: Differential (*Edge − Corner*) in edge-degradation accuracy relative to corner-degradation accuracy on the shape and image recovery tasks. As $p_d$ increases, humans retain high accuracy on edge-degraded images but perform worse on corner-degraded images. On the other hand, MLP-Mixer quickly experiences decreasing accuracy on edge-degraded images, while retaining high accuracy on corner-degraded images. Overall, MLP-Mixer and human subjects exhibit starkly contrasting behavior.

performance for ResNet-18. While the general trend of these models is the same as their ImageNet counterparts, these models retain greater accuracy on corner-degraded shapes and fail more rapidly

| $p_d$ and Type | Triangle | Square | Pentagon | Hexagon | Heptagon | Octagon |
|---|---|---|---|---|---|---|
| 0.3 Edge Degradation | 100.0 | 67.7 | 92.2 | 0.2 | 0.9 | 100.0 |
| 0.4 Edge Degradation | 100.0 | 27.6 | 0.0 | 0.0 | 0.0 | 94.7 |
| 0.5 Edge Degradation | 100.0 | 39.0 | 0.0 | 0.0 | 0.1 | 92.9 |
| 0.6 Edge Degradation | 100.0 | 32.7 | 0.0 | 0.0 | 0.0 | 0.1 |
| 0.7 Edge Degradation | 6.6 | 98.9 | 0.0 | 0.0 | 0.0 | 0.0 |
| 0.3 Corner Degradation | 100.0 | 96.0 | 88.9 | 87.0 | 93.7 | 82.3 |
| 0.4 Corner Degradation | 100.0 | 79.2 | 55.9 | 66.3 | 42.4 | 60.5 |
| 0.5 Corner Degradation | 100.0 | 69.8 | 0.1 | 14.4 | 16.3 | 55.6 |
| 0.6 Corner Degradation | 100.0 | 16.1 | 0.0 | 0.0 | 0.0 | 0.0 |
| 0.7 Corner Degradation | 60.4 | 95.2 | 0.0 | 0.0 | 0.0 | 0.0 |

Table 1: Per-class accuracy of ImageNet-pretrained ResNet-18 at varying degradation proportions.

| $p_d$ and Type | Triangle | Square | Pentagon | Hexagon | Heptagon | Octagon |
|---|---|---|---|---|---|---|
| 0.3 Edge Degradation | 100.0 | 99.3 | 98.6 | 30.1 | 9.4 | 100.0 |
| 0.4 Edge Degradation | 97.5 | 98.3 | 89.5 | 1.7 | 0.0 | 100.0 |
| 0.5 Edge Degradation | 58.0 | 57.0 | 2.4 | 0.2 | 0.2 | 99.9 |
| 0.6 Edge Degradation | 2.4 | 27.0 | 0.4 | 0.0 | 0.1 | 100.0 |
| 0.7 Edge Degradation | 0.0 | 12.0 | 0.0 | 0.1 | 69.7 | 26.5 |
| 0.3 Corner Degradation | 100.0 | 100.0 | 100.0 | 100.0 | 11.4 | 0.0 |
| 0.4 Corner Degradation | 100.0 | 100.0 | 100.0 | 100.0 | 99.1 | 0.0 |
| 0.5 Corner Degradation | 100.0 | 100.0 | 100.0 | 100.0 | 32.3 | 0.1 |
| 0.6 Corner Degradation | 100.0 | 98.5 | 100.0 | 62.5 | 100.0 | 84.0 |
| 0.7 Corner Degradation | 99.9 | 89.9 | 53.5 | 99.8 | 84.6 | 0.0 |

Table 2: Per-class accuracy of FractalDB-pretrained ResNet-18 at varying degradation proportions.

on edge-degraded shapes as degradation proportion increases, diverging even further from human behavior.

For fractals generated by linear Iterated Function Systems (IFS), such as those in FractalDB, the resulting images primarily exhibit the NAPs of collinearity and symmetry, and not so much structurally complex vertices. Therefore, it is not surprising that these fractal-pretrained models are more amenable to processing edges, thus performing even better on corner-degraded shapes. In that sense, fractal pretraining further misaligns neural network behavior from humans, raising the question of if standard fractal pretraining can indeed be a suitable substitute for natural image pretraining. However, we note that extensions to the traditional IFS generative process exist that enable emphasis on different NAPs. For example, the Fractal Flame algorithm (Spotworks and Berthoud, 2008) produces *curved* fractals, thus emphasizing curvilinearity. We leave the investigation of such fractal generation procedures' effects on neural network behavior and alignment for future work.

**Visualizing Network Internals**    To gain further intuition for neural network behavior, we study our learned ResNet-18 models by analyzing their corresponding Grad-CAM visualizations (Selvaraju et al., 2017). Figure 7 shows an illustrative example of the difference in gradients for the *true* target concept flowing into the final convolutional layer between ImageNet-pretrained ResNet-18 and FractalDB-pretrained ResNet-18. On two randomly selected pentagons from our dataset and their corresponding 50% edge-degraded and 50% corner-degraded variants, the highlighted regions of the ImageNet model are less concentrated within the shape's radius than the highlighted regions of the FractalDB model, suggesting that fractal pretraining endowed ResNet-18 with a somewhat more robust ability to classify degraded shapes. Though not a full explanation for model behavior, we see such discrepancies as a potential indication that fractal pretraining enables models to learn and leverage more robust geometric features, though not necessarily human-like vertex features.

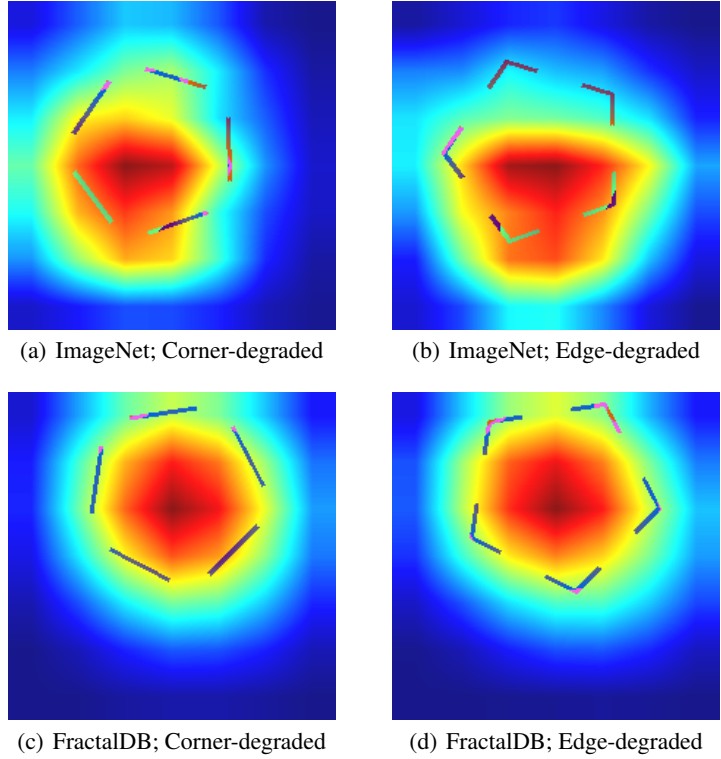



(a) ImageNet; Corner-degraded      (b) ImageNet; Edge-degraded

(c) FractalDB; Corner-degraded      (d) FractalDB; Edge-degraded



Figure 7: Grad-CAM visualizations with respect to the true pentagon class for ResNet-18 on (a) 50% corner-degraded pentagon with an ImageNet-pretrained model, (b) 50% edge-degraded pentagon with an ImageNet-pretrained model, (c) 50% corner-degraded pentagon with a FractalDB-pretrained model, and (d) 50% edge-degraded pentagon with a FractalDB-pretrained model.

## 5 Conclusion

Inspired by cognitive science and the theory of Recognition-by-Components, we introduce a new perspective on the human-machine vision gap and investigate the notion of image recovery in degraded regular polygons. To do so, we develop the Automated Shape Recoverablility Test pipeline for generating regular polygon sketches at scale with varying orders of degradation, which we open-source to the machine learning community. We then train common deep learning vision models for this simple task and find that their behavior fundamentally conflicts with how humans perceive images. Furthermore, we show that fractal pretraining further misaligns neural network and human behavior. Based on these results, we encourage further investigation into this fundamental conflict between human and machine vision.

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
