# DATASHEET:
# Recognition-by-Components Degraded Polygons

**Leonard Tang**
**Harvard University**
leonardtang@college.harvard.edu

**Dan Ley**
**Harvard University**
dley@g.harvard.edu

## MOTIVATION

**For what purpose was the dataset created?** Was there a specific task in mind? Was there a specific gap that needed to be filled? Please provide a description.
This dataset was created in order to study modern deep learning vision systems from a principled, cognitive science perspective. In particular, we aim to study human and machine vision misalignment by analyzing their behavior on the *image recovery* task. We revisit and modernize the theory of Recognition-by-Components by generating this dataset.

**Who created this dataset (e.g., which team, research group) and on behalf of which entity (e.g., company, institution, organization)?**
Leonard Tang and Dan Ley from Harvard University created this dataset.

**What support was needed to make this dataset?** (e.g.who funded the creation of the dataset? If there is an associated grant, provide the name of the grantor and the grant name and number, or if it was supported by a company or government agency, give those details.)
No funding was necessary to create this dataset.

**Any other comments?**
We also note that we do not provide just a dataset, but also an algorithm for synthetically generating a (theoretically) unbounded number of datapoints.

## COMPOSITION

**What do the instances that comprise the dataset represent (e.g., documents, photos, people, countries)?** Are there multiple types of instances (e.g., movies, users, and ratings; people and interactions between them; nodes and edges)? Please provide a description.
The instances in the dataset are black-on-white *polygon sketches*, similar to those tested in Recognition-by-Components theory. Each instance exhibits a varying amount of perimeter degradation, either at the corners or edges.

**How many instances are there in total (of each type, if appropriate)?**
We produce 1,260,000 sketches of regular polygons evenly distributed across 7 shape categories, 9 levels of image degradation, and 2 forms of degradation (corner and edge degradation).

**Does the dataset contain all possible instances or is it a sample (not necessarily random) of instances from a larger set?** If the dataset is a sample, then what is the larger set? Is the sample representative of the larger set (e.g., geographic coverage)? If so, please describe how this representativeness was validated/verified. If it is not representative of the larger set, please describe why not (e.g., to cover a more diverse range of instances, because instances were withheld

or unavailable).

As mentioned previously, there are an infinite number of possible instances that can be generated using our pipeline. However, we believe that the dataset we create is sufficiently broad and varied (according to the stochasticity of our generation procedure), densely covering the space of all possible instances.

**What data does each instance consist of?** "Raw" data (e.g., unprocessed text or images) or features? In either case, please provide a description.
Each instance is a PNG, with CSV files describing metadata for each PNG, which provides a quick way to access/bookkeep the data.

**Is there a label or target associated with each instance?** If so, please provide a description.
Yes, the label of each instance is the category of shape (which $n$-gon) it belongs too.

**Is any information missing from individual instances?** If so, please provide a description, explaining why this information is missing (e.g., because it was unavailable). This does not include intentionally removed information, but might include, e.g., redacted text.
No.

**Are relationships between individual instances made explicit (e.g., users' movie ratings, social network links)?** If so, please describe how these relationships are made explicit.
We link the original, whole, version of a polygon with its edge-degraded and corner-degraded counterparts through the index naming of each PNG, and its corresponding location in the CSV metadata.

**Are there recommended data splits (e.g., training, development/validation, testing)?** If so, please provide a description of these splits, explaining the rationale behind them.
We provide the suggested data splits in the main paper.

**Are there any errors, sources of noise, or redundancies in the dataset?** If so, please provide a description.
N/A

**Is the dataset self-contained, or does it link to or otherwise rely on external resources (e.g., websites, tweets, other datasets)?** If it links to or relies on external resources, a) are there guarantees that they will exist, and remain constant, over time; b) are there official archival versions of the complete dataset (i.e., including the external resources as they existed at the time the dataset was created); c) are there any restrictions (e.g., licenses, fees) associated with any of the external resources that might apply to a future user? Please provide descriptions of all external resources and any restrictions associated with them, as well as links or other access points, as appropriate.
Yes, it is self-contained.

**Does the dataset contain data that might be considered confidential (e.g., data that is protected by legal privilege or by doctor-patient confidentiality, data that includes the content of individuals' non-public communications)?** If so, please provide a description.
No.

**Does the dataset contain data that, if viewed directly, might be offensive, insulting, threatening, or might otherwise cause anxiety?** If so, please describe why.
No.

**Does the dataset relate to people?** If not, you may skip the remaining questions in this section.
No.

**Does the dataset identify any subpopulations (e.g., by age, gender)?** If so, please describe how these subpopulations are identified and provide a description of their respective distributions within the dataset.
N/A

**Is it possible to identify individuals (i.e., one or more natural persons), either directly or indirectly (i.e., in combination with other data) from the dataset?** If so, please describe how.
N/A

**Does the dataset contain data that might be considered sensitive in any way (e.g., data that reveals racial or ethnic origins, sexual orientations, religious beliefs, political opinions or union memberships, or locations; financial or health data; biometric or genetic data; forms of government identification, such as social security numbers; criminal history)?** If so, please provide a description.
N/A

**Any other comments?**
N/A

## COLLECTION

**How was the data associated with each instance acquired?** Was the data directly observable (e.g., raw text, movie ratings), reported by subjects (e.g., survey responses), or indirectly inferred/derived from other data (e.g., part-of-speech tags, model-based guesses for age or language)? If data was reported by subjects or indirectly inferred/derived from other data, was the data validated/verified? If so, please describe how.
We procedurally and stochastically generate our dataset according to decades-old cognitive science theory and experiments.

**Over what timeframe was the data collected?** Does this timeframe match the creation timeframe of the data associated with the instances (e.g., recent crawl of old news articles)? If not, please describe the timeframe in which the data associated with the instances was created. Finally, list when the dataset was first published.
The dataset was generated in under an hour (not including developer time to produce the generation code). The dataset was first published on June 7, 2023.

**What mechanisms or procedures were used to collect the data (e.g., hardware apparatus or sensor, manual human curation, software program, software API)?** How were these mechanisms or procedures validated?
Software was used to produce the dataset. Please see our repository for more context.

**What was the resource cost of collecting the data?** (e.g. what were the required computational resources, and the associated financial costs, and energy consumption - estimate the carbon footprint. See Strubell *et al.*? for approaches in this area.)
Despite the size of the dataset, its computation cost is relatively low. For example, our pipeline requires less than 30 seconds of compute time to serially generate 6,000 diverse images across 7 shape categories – triangles, squares, pentagons, hexagons, heptagons, and octagons – with 1,000 images each (benchmarked on a M1 chip).

**If the dataset is a sample from a larger set, what was the sampling strategy (e.g., deterministic, probabilistic with specific sampling probabilities)?**
Our actual dataset is not sampled. For experiments, we sample uniformly across our dataset.

**Who was involved in the data collection process (e.g., students, crowdworkers, contractors) and how were they compensated (e.g., how much were crowdworkers paid)?**
No manual labor was involved in creating this dataset.

**Were any ethical review processes conducted (e.g., by an institutional review board)?** If so, please provide a description of these review processes, including the outcomes, as well as a link or other access point to any supporting documentation.
N/A

**Does the dataset relate to people?** If not, you may skip the remainder of the questions in this section.
N/A

**Did you collect the data from the individuals in question directly, or obtain it via third parties or other sources (e.g., websites)?**
N/A

**Were the individuals in question notified about the data collection?** If so, please describe (or show with screenshots or other information) how notice was provided, and provide a link or other access point to, or otherwise reproduce, the exact language of the notification itself.
N/A

**Did the individuals in question consent to the collection and use of their data?** If so, please describe (or show with screenshots or other information) how consent was requested and provided, and provide a link or other access point to, or otherwise reproduce, the exact language to which the individuals consented.
N/A

**If consent was obtained, were the consenting individuals provided with a mechanism to revoke their consent in the future or for certain uses?** If so, please provide a description, as well as a link or other access point to the mechanism (if appropriate)
N/A

**Has an analysis of the potential impact of the dataset and its use on data subjects (e.g., a data protection impact analysis)been conducted?** If so, please provide a description of this analysis, including the outcomes, as well as a link or other access point to any supporting documentation.
N/A

**Any other comments?**
N/A

## PREPROCESSING / CLEANING / LABELING

**Was any preprocessing/cleaning/labeling of the data done(e.g.,discretization or bucketing, tokenization, part-of-speech tagging, SIFT feature extraction, removal of instances, processing of missing values)?** If so, please provide a description. If not, you may skip the remainder of the questions in this section.
N/A

**Was the "raw" data saved in addition to the preprocessed/cleaned/labeled data (e.g., to support unanticipated future uses)?** If so, please provide a link or other access point to the "raw" data.
N/A

**Is the software used to preprocess/clean/label the instances available?** If so, please provide a link or other access point.

N/A

**Any other comments?**

N/A

## USES

**Has the dataset been used for any tasks already?** If so, please provide a description.

Yes, we use our dataset to perform a comprehensive analysis of various neural network architectures on the task of shape recovery across varying levels of image degradation. We demonstrate a striking discrepancy in how machine learning models and humans perceive images. Unlike humans, neural networks consistently rely more on edges than corners for image recovery, pointing to a fundamental difference in image processing between machines and humans.

**Is there a repository that links to any or all papers or systems that use the dataset?** If so, please provide a link or other access point.

Yes, the link to our repository is here.

**What (other) tasks could the dataset be used for?**

We envision other researchers using this dataset to study the fundamental science and perceptual behavior of neural networks.

**Is there anything about the composition of the dataset or the way it was collected and preprocessed/cleaned/labeled that might impact future uses?** For example, is there anything that a future user might need to know to avoid uses that could result in unfair treatment of individuals or groups (e.g., stereotyping, quality of service issues) or other undesirable harms (e.g., financial harms, legal risks) If so, please provide a description. Is there anything a future user could do to mitigate these undesirable harms?

There is no such bias in the dataset.

**Are there tasks for which the dataset should not be used?** If so, please provide a description.

As long as the task is in the vision domain, we feel that this dataset is relevant.

**Any other comments?**

N/A

## DISTRIBUTION

**Will the dataset be distributed to third parties outside of the entity (e.g., company, institution, organization) on behalf of which the dataset was created?** If so, please provide a description.

It will be open sourced to individuals.

**How will the dataset will be distributed (e.g., tarball on website, API, GitHub)?** Does the dataset have a digital object identifier (DOI)?

The dataset is available on GitHub at this link.

**When will the dataset be distributed?**

It is already distributed.

**Will the dataset be distributed under a copyright or other intellectual property (IP) license, and/or under applicable terms of use (ToU)?** If so, please describe this license and/or ToU, and provide a link or other access point to, or otherwise reproduce, any relevant licensing terms or ToU, as well as any fees associated with these restrictions.
Our dataset is available under the MIT License.

**Have any third parties imposed IP-based or other restrictions on the data associated with the instances?** If so, please describe these restrictions, and provide a link or other access point to, or otherwise reproduce, any relevant licensing terms, as well as any fees associated with these restrictions.
No.

**Do any export controls or other regulatory restrictions apply to the dataset or to individual instances?** If so, please describe these restrictions, and provide a link or other access point to, or otherwise reproduce, any supporting documentation.
No.

**Any other comments?**
N/A

---

## MAINTENANCE

**Who is supporting/hosting/maintaining the dataset?**
Leonard Tang is maintaining this dataset.

**How can the owner/curator/manager of the dataset be contacted (e.g., email address)?**
You can reach Leonard at leonardtang@college.harvard.edu.

**Is there an erratum?** If so, please provide a link or other access point.
No.

**Will the dataset be updated (e.g., to correct labeling errors, add new instances, delete instances)?** If so, please describe how often, by whom, and how updates will be communicated to users (e.g., mailing list, GitHub)?
We will handle all updates via GitHub, responding promptly to user issues.

**If the dataset relates to people, are there applicable limits on the retention of the data associated with the instances (e.g., were individuals in question told that their data would be retained for a fixed period of time and then deleted)?** If so, please describe these limits and explain how they will be enforced.
N/A

**Will older versions of the dataset continue to be supported/hosted/maintained?** If so, please describe how. If not, please describe how its obsolescence will be communicated to users.
Yes. We will host old versions of the dataset on GDrive.

**If others want to extend/augment/build on/contribute to the dataset, is there a mechanism for them to do so?** If so, please provide a description. Will these contributions be validated/verified? If so, please describe how. If not, why not? Is there a process for communicating/distributing these contributions to other users? If so, please provide a description.
Yes. Since we are using GitHub, users are free to raise issues and introduce changes via Pull Requests. We encourage user discussion and activity to refine this dataset.

**Any other comments?**
N/A

# 1 Author Statement

We hereby state that we bear all responsibility in case of violation of rights and misuse of this dataset. We also confirmation the data license.