# OpenReview forum: "Degraded Polygons Raise Fundamental Questions of Neural Network Perception"
_NeurIPS.cc/2023/Track/Datasets_and_Benchmarks — NeurIPS 2023 Datasets and Benchmarks Poster_

### Official Review · Reviewer_5Kr2 · 2023-07-20
**Initial review**

**Rating:** 9
**Confidence:** 3
**Correctness:** The methods and data described here s…

**Strengths:**

The striking difference in human- and machine-learned perceptions of such simple images is one of the most interesting aspects of this work. The authors have come up with an intuitive benchmark dataset that seems very promising to help improve our understanding of how image-based models make their decisions.

**Additional Feedback:**

Overall, I think the work is very interesting! I have a few questions/suggestions, but some of them probably make the most sense to address in followup work.

1. I might suggest separating the full set of generated images into their own repository.
2. I find the non-monotonic performance in Tables 1 and 2 for higher-symmetry shapes, particularly octagons, to be very intriguing; do the authors have any idea why octagons seem to be so much easier to identify, or if this may be some artifact of the data preparation process?
3. Testing on architectures that are commonly used makes perfect sense, but I wonder if shallower, simpler (more toy-grade) CNN architectures might place more emphasis on corners in their prediction; intuitively, a single patch of pixels that covers a corner should be enough to solve the classification problem, while you must span enough pixels to cover two lines if the corners are missing.

**Clarity:**

Overall, the paper is clear. Perhaps I missed it, but a more thorough description of the fractal pretraining (in terms of what the network is being trained on: is it learning to classify types of fractals? Are patches being masked out to be filled in by the network?) would be helpful.

**Documentation:**

The data generation process seems to be well-described in the linked github repository.

**Limitations:**

The authors have sufficiently addressed the limitations of their work in my opinion.

**Opportunities For Improvement:**

It would be good to incorporate error bars (or some other statement of reproducibility, for example, if the error bars are always smaller than the plot markers themselves) into the plots, even if they are negligibly small, as the authors state in the checklist; "not sufficient variation" could mean different things to different people, and one of the most interesting plots to me (the crossover in the ViT result of Figure 3) seems like it could be impacted by small changes in the results.

**Relation To Prior Work:**

The paper describes its place in the landscape of prior work well.

**Summary And Contributions:**

This paper describes a simple computer vision benchmark dataset generated by removing edges or corners from regular polygons and training standard image classifiers to identify the symmetry of the shapes. Intriguingly, the authors find that trained networks usually perform better when removing corners of shapes than when edges are erased; this is in contrast to how humans tend to perform. I think that the simplicity of the approach and interesting discrepancy between human and machine learned results combine to present some very compelling initial insights.

---

> ### Author Response · Authors · 2023-08-21
>
> We thank the reviewer for their interest in our work and insightful comments!
>
> > **It would be good to incorporate error bars (or some other statement of reproducibility, for example, if the error bars are always smaller than the plot markers themselves) into the plots, even if they are negligibly small, as the authors state in the checklist; "not sufficient variation" could mean different things to different people, and one of the most interesting plots to me (the crossover in the ViT result of Figure 3) seems like it could be impacted by small changes in the results.**
>
> Thank you for the suggestion! We have obtained these results and are in the process of incorporating them into the revision.
>
> > **Overall, the paper is clear. Perhaps I missed it, but a more thorough description of the fractal pretraining (in terms of what the network is being trained on: is it learning to classify types of fractals? Are patches being masked out to be filled in by the network?) would be helpful.**
>
> We apologize for the confusion regarding fractal pretraining. To be more explicit, fractal pretraining relies on a concept called Formula-driven Supervised Learning (FDSL), in which fractals are generated by mathematical formulae, then binned into categories based on the range of formula parameters. The learning task is thus to match generated fractals with the correct class (i.e. range of parameters). We have updated the Related Work to reflect this.
>
> > **I might suggest separating the full set of generated images into their own repository.**
>
> We thank the reviewer for their suggestion. However, because the storage size of the full dataset is relatively manageable, we believe it best to pair the data with the training and analysis code rather than separate the two. That being said, we will make a link that standalone hosts the final dataset for further convenience.
>
> > **I find the non-monotonic performance in Tables 1 and 2 for higher-symmetry shapes, particularly octagons, to be very intriguing; do the authors have any idea why octagons seem to be so much easier to identify, or if this may be some artifact of the data preparation process?**
>
> We are equally intrigued by this observation as well; we had performed initial analysis in this setting, but decided against including it in this particular work, as we felt that it would detract from the main discrepancy between using edge and corner features. We are actively and excitedly pursuing this line of research for future work!
>
> > **Testing on architectures that are commonly used makes perfect sense, but I wonder if shallower, simpler (more toy-grade) CNN architectures might place more emphasis on corners in their prediction; intuitively, a single patch of pixels that covers a corner should be enough to solve the classification problem, while you must span enough pixels to cover two lines if the corners are missing.**
>
> We thank the reviewer for their suggestion. We had also considered using simple architectures for this task, but ultimately decided against studying them for this work. We envision these simple architectures as a core part of future work that searches for the “right” architectures to align human and machine vision. In this work, we sought primarily to lay down the foundations and benchmarks for discussing and investigating a particular facet of human-machine misalignment. In future work, we are excited to develop novel architectures, likely starting from simpler CNN architectures as the reviewer suggested, that are able to handle this misalignment while balancing general performance.

---

### Official Review · Reviewer_fh2w · 2023-07-20

**Rating:** 7
**Confidence:** 4

**Strengths:**

The paper presents a very simply but also very intriguing property of CNNs, which has not been described before but also enriches the broughter current discussion on model biases. The experimental setup is well motivated from prior work in other research communities and has the potential to introduce new perspectives and methods into the ML comunity.

**Additional Feedback:**

none

**Clarity:**

The paper is well written, organized and motivated. The only point missing would be to show some examples of the actual data (which is not even in the supplemental material). Here I would suggest to give up some space from fig 2, which could be moved to an appendix in favour of some data examples, including samples where CNNs would fail but humans not (fig 1 only shows an illustration, not real data).



**Correctness:**

The introduce methods for data generation and evaluation are sound. However, I have some doubts about validity of the Grad-Cam experiments. First, results from attribution methods (not only gradCam) are not independent from their parameters (which are neither reported nor changed). Recent works also shead strong doubts on the over all validity and stability of CAM methods, hence I would argue to discard this part in favor of the other points mentioned above...




**Documentation:**

the data set and generation process are well documented and appear to be easy to access and reproduce.

**Ethics:**

no ethical concerns here

**Limitations:**

The presented data (and problem) are interesting, but very basic. It remains largely unclear how the proposed setup will translate to real world classification problems, whereas the existing dimensions like robustness and texture bias have effect that  can be observed in real world applications. Here the paper somewhat fails to at least show a perspective...

**Opportunities For Improvement:**

Reading the paper raises many followup questions, which of cause can not all be answered in a single paper. However, the paper would benefit strongly if it at least would provide some outlook into more complex, more realistic scenarios - maybe in object segmentation or a a setup with textured objects.

Also I would find it  very interesting to some confusion-matrices for the experiments: which class is confused with which under degradation? Does this also differ between humans and CNNs?

The next question would be, if analog to adversarial training, models could be made more "human like" if trained on additional data.  [1] Showed this effect for the texture bias.

[1] Gavrikov, P., Keuper, J., & Keuper, M. (2023). An Extended Study of Human-like Behavior under Adversarial Training. In Proceedings of the IEEE/CVF Conference on Computer Vision and Pattern Recognition (pp. 2360-2367).

**Relation To Prior Work:**

The prior work, especially the out of community work is discussed at the right level and gives a good background to the proposed dataset, methods  and evaluation.

**Summary And Contributions:**

The  paper introduces a novel and interesting perspective on the misalignment between current deep learning models (CNNs in this specific case) and human recognition abilities. The proposed shape degradation recovery property adds a new dimension to the ongoing lack of robustness and texture bias discussion.  To this end, the authors present a dataset and generator framework for seemingly simple task of classifying binary shape contours, under the test degradation of edges or corners. The provided experiments show, that CNNs not only mostly fail to prevail in this task, but are also using different shape features than humans (mostly focusing on edes instead of corners).

---

> ### Comment · Reviewer_fh2w · 2023-08-21
> **Revise paper and respond to positive reviews**
>
> Dear authors,
>
> it appears that you only answered the negative reviews so far.  I would advise you to also answer to the positive reviews (we still had some minor points) and to take the opportunity  revise the paper and summarize the changes (citation from the official author rebuttal email):
>
> * *You can respond to each review with a separate response, and also have a “global” response to all reviewers jointly. Use the “Official Comment” button to submit these.*
>
> * *You can make revisions to your paper and supplementary materials, and you are allowed one additional page to address the reviewers’ comments. Ensure that it is easy for reviewers to find how their comments were addressed.*
>
> This would make it much easier to argue in favor of you submission in the upcoming discussions...

---

> > ### Author Response · Authors · 2023-08-21
> >
> > Thank you for the message! We apologize for the delay in response -- we had decided to address the reviews in the order in which they are listed. We thank the reviewer for their interest in and feedback on our work!
> >
> > > **Reading the paper raises many followup questions, which of course can not all be answered in a single paper. However, the paper would benefit strongly if it at least would provide some outlook into more complex, more realistic scenarios - maybe in object segmentation or a setup with textured objects.**
> >
> > See response to limitations.
> >
> > > **Also I would find it very interesting to some confusion-matrices for the experiments: which class is confused with which under degradation? Does this also differ between humans and CNNs?**
> >
> > Thank you for the suggestion! We have obtained these results and are in the process of incorporating them into the revision.
> >
> > > **The presented data (and problem) are interesting, but very basic. It remains largely unclear how the proposed setup will translate to real world classification problems, whereas the existing dimensions like robustness and texture bias have effect that can be observed in real world applications. Here the paper somewhat fails to at least show a perspective...**
> >
> > While we understand the reviewer’s perspective and they propose valid potential, additional experimentation scenarios, we believe that these squarely belong in future work, as they would otherwise muddle the main contribution of this work. We re-emphasize the fact that “the simple structure of our data, being merely regular polygons, indicates a _**fundamental misalignment**_ of deep learning vision models with respect to human vision,” as stated in our paper. Moreover, we view our dataset as a minimum viable task that can induce and highlight this misalignment, which is also noted in our paper. We have revised our writing to highlight this perspective more explicitly in our introduction and abstract.
> >
> > > **The introduce methods for data generation and evaluation are sound. However, I have some doubts about validity of the Grad-Cam experiments. First, results from attribution methods (not only gradCam) are not independent from their parameters (which are neither reported nor changed). Recent works also shead strong doubts on the over all validity and stability of CAM methods, hence I would argue to discard this part in favor of the other points mentioned above…**
> >
> > We thank the reviewer for bringing up these important points. To that end, we have included the model checkpoints (i.e. parameters) that we used for performing our Grad-CAM analyses in the repository. To the second point regarding the validity of Grad-CAM, concerns generally center on 1) multiple occurrences of objects in an image and 2) object localization within complex backgrounds. As our images fall into neither of these categories, Grad-CAM remains a valid method for exploring our models’ internals.
> >
> > > **The paper is well written, organized and motivated. The only point missing would be to show some examples of the actual data (which is not even in the supplemental material). Here I would suggest to give up some space from fig 2, which could be moved to an appendix in favour of some data examples, including samples where CNNs would fail but humans not (fig 1 only shows an illustration, not real data).**
> >
> > We apologize for the confusion regarding the presentation of our data. To be clear, the degraded shapes included in Figure 1 are indeed real examples pulled directly from our dataset, and in particular examples for which CNNs fail/succeed and humans succeed/fail respectively. We have revised the figure caption to make this clearer.

---

### Official Review · Reviewer_Q8xH · 2023-07-21
**Interesting results. But significance and generality of the findings are unclear.**

**Rating:** 4
**Confidence:** 3
**Correctness:** Yes
**Clarity:** Yes, very well written.Yes

**Strengths:**

1.	Studies like this comparing network properties to characteristic properties of humans (found in perception or cognitive science literature) are important to keep track of the similarities and differences between human and network behavior.
2.	Experiments use standard networks and training settings.
3.	Code and dataset are public.


**Additional Feedback:**

N/A

**Documentation:**

Yes

**Limitations:**

Yes

**Opportunities For Improvement:**

1.	Although, as I said in strength #1, work comparing humans and networks is useful, the significance and generality of the comparison in this paper is not clear to me. Why are the findings that people rely on corners and networks rely on edges important for machine learning research? In the introduction, the paper references two popularly found divergences between human and network behavior – adversarial robustness and texture-vs-shape bias. Adversarial robustness is a general finding that is important for AI because networks used for real-world applications can be susceptible to attacks that don’t affect humans. Texture-vs-shape bias is also a general finding, found in many types of images and datasets (natural images, letters etc.). The results in this paper are specific to outlines of regular polygons. Why are polygons important for vision in general? And how can the experiments in this paper be extended to more real-world tasks such as ImageNet?
As the paper mentions, the human-network comparison is not a controlled experiment since the human results haven’t been reproduced on the same task that used to evaluate networks. Additionally, the paper refers to the generated images as “degraded” polygons, but the only kind of degradation used is occlusion. These issues add on to my concern about generalizability of the experiments and results.
2.	I felt that the GradCAM analysis does not reveal any new information that the previous sections haven’t already found. Could the authors clarify why this analysis is important?
3.	The paper covers the inspiration from Recognition-by-components theory in cognitive science very well and in much detail. I would urge the authors to also look at the large Gestalt literature in visual perception. Specifically, the notion of Gestalt closure connects very strongly to the issues pointed out in this paper. For recent work evaluating networks on Gestalt closure, see Kim et al. 2019.
4.	Figure 5 states that networks pretrained on Fractals “Compared to their ImageNet-pretrained counteparts, however, ResNet-18 and ResNet-50 both retain performance better on edge-degraded shapes”. From the figure, this seems like a typo. Shouldn’t it instead be:  “ResNet-18 and ResNet-50 both retain performance better on corner-degraded shapes”?

Reference:

Kim, B., et al. "Neural networks trained on natural scenes exhibit gestalt closure. arXiv." arXiv preprint arXiv:1903.01069 (2019).


**Relation To Prior Work:**

Yes

**Summary And Contributions:**

Inspired by shape recoverability literature in cognitive science, this paper develops an “automated shape recoverability test”, a method to quickly generate a large dataset of whole and degraded regular polygons which the authors use to evaluate neural networks. They find that contrary to humans, networks pretrained on ImageNet are more affected by edge-degradations than corner-degradations. Pretraining with fractals instead of ImageNet only increases the human-network disparity. GradCAM-analysis confirms this observation.

---

> ### Author Response · Authors · 2023-08-19
>
> We thank the reviewer for the careful response, and appreciate their insightful comments.
>
> > **Although, as I said in strength #1, work comparing humans and networks is useful, the significance and generality of the comparison in this paper is not clear to me. Why are the findings that people rely on corners and networks rely on edges important for machine learning research?**
>
> We believe that the fundamental disparities we demonstrate between human and machine vision (namely, that humans rely on corners and that neural networks rely on edges) are of critical interest to the community. Given the importance of AI alignment, understanding and correcting for misaligned model behavior on our benchmark – which is an *intentionally simple instantiation* of a dataset that induces human-machine misalignment – is a worthwhile and necessary endeavor.
>
> If machine learning researchers cannot even achieve alignment on our dataset (in the form of using human-like usage of corner and edge features for shape recovery), then clearly there exists a gaping flaw in the current state of research and approaches to computer vision.
>
> > **The results in this paper are specific to outlines of regular polygons. Why are polygons important for vision in general?**
>
> Firstly, shape primitives have long been proposed as components for object models in the visual system from a cognitive science perspective (e.g. Shams and Malsburg, 2002). Thus, we believe that studying neural networks’ behavior on variants of such primitives yields a valuable comparison with the human visual system.
>
> Secondly, we believe that it is prudent to study simple but abnormal phenomena in neural networks. Many researchers focus immediately on applying deep learning to real-world settings without pausing to duly reflect on underlying characteristics of their system. Until we understand from a principled perspective – e.g. via tasks such as ours – how and why neural networks exhibit certain core properties, we will inevitably uncover new examples of these models’ brittleness, such as adversarial attacks and texture bias. That is, rather than patch these “bugs” at the surface level, we advocate for investigating and eliminating them at their root.
>
> > **And how can the experiments in this paper be extended to more real-world tasks such as ImageNet?**
>
> Please see above.
>
> > **Additionally, the paper refers to the generated images as “degraded” polygons, but the only kind of degradation used is occlusion. These issues add on to my concern about generalizability of the experiments and results.**
>
> We apologize for the confusion regarding the term “degraded.” In cognitive science and Recognition-by-Components theory, the term “degraded” refers precisely to the deletion of contours that we study here. To produce the fairest comparison, we therefore remain faithful to the original human setting when benchmarking machine capabilities.
>
> > **I felt that the GradCAM analysis does not reveal any new information that the previous sections haven’t already found. Could the authors clarify why this analysis is important?**
>
> While other sections in our work address the discrepancies between the behavior of ImageNet- and FractalDB-pretrained models from a pure results perspective, our GradCAM analysis provides a deeper initial look into how these models’ recognition processes differ. To be precise, it is not immediately clear from our FractalDB model results (i.e. Figure 5 and Table 2) that these models have learned a more robust ability to leverage geometric features for classifying degraded shapes, which is what our GradCAM analysis alludes to.
>
> > **The paper covers the inspiration from Recognition-by-components theory in cognitive science very well and in much detail. I would urge the authors to also look at the large Gestalt literature in visual perception. Specifically, the notion of Gestalt closure connects very strongly to the issues pointed out in this paper. For recent work evaluating networks on Gestalt closure, see [Kim et al. 2019].**
>
> In general, we view the notion of image recoverability as subtly, but fundamentally, different from the principle of Gestalt closure. In particular, Gestalt closure posits that incomplete visual stimuli are perceived by humans as a single, continuous object, where this underlying object is the single source of truth. However, in RBC and image recoverability, degraded stimuli may not even be able to be recovered at all – it is possible that there is an ambiguous one-to-many mapping between the degraded stimuli and the original object. For a concrete example, consider a square with corner degradation. Perceiving the degraded stimuli alone, it is unclear what the underlying recovered object should be – it could have indeed been a square, but it could have also easily been a hexagon, octagon, and so on. Regardless, we have included the reviewer’s reference in the updated version.

---

### Official Review · Reviewer_d9g1 · 2023-07-22
**Interesting observations, but seem not to be very relevant to datasets and benchmarks track**

**Rating:** 6
**Confidence:** 3
**Clarity:** Reasonable.

**Strengths:**

The paper is well written, the presentation is quite logical and easy to follow. The experiments are convincing and support the claims about neural networks' perception. The obtained experimental result is interesting.

**Additional Feedback:**

No

**Correctness:**

Bitterman (1987) seems to be the only reference supporting the claim that humans better rely on corners than edges. This is not super convincing and I recommend to find more citations here.

**Documentation:**

Good enough.

**Limitations:**

Potential impact of the results is not very clear (see opportunities for improvement).

**Irrelevance to the datasets and benchmark track (?)** Overall, I do not dot understand why this paper is submitted to the benchmark and datasets track. To be precise, the dataset created here is rather simple (and, in my opinion, this dataset is not a significant contribution) and it seems to me that the main contribution is their empirical findings about neural networks. Hence the current paper may better suit the main conference track or, even better, some other conference or journal, possibly related to human/AI perception etc.

**Opportunities For Improvement:**

Although the obtained result interesting, It is unclear whether it is of any use for machine learners/computer vision algorithms developers. i think the authors should provide a discussion how their fundamental findings can be potentially used in practice.

**Relation To Prior Work:**

I am not an expert in this field of neural networks perception and can not carefully access this aspect.

**Summary And Contributions:**

The paper studies the behavior of neural networks in classifying images under degradation. For this, they create a synthetic dataset of polygons with various degradations and use it test how accurately various neural network architectures classify these polygons. The main interesting observation is that neural networks focus on different aspects than humans (rely more on edges than corners). This is an interesting discrepancy between the humans and neural networks perception. In order to further support their findings about the neural networks perception, the authors conduct an extra experiment with fine-tuning the models pre-trained on fractals and observe analogous things: neural networks work better when they they see edges than corners.

**Update:** 4->6, see the discussion below

---

> ### Author Response · Authors · 2023-08-19
>
> We thank the reviewer for their response! We are glad to see that you approve of the paper’s presentation, and find the experimental results both convincing and interesting.
>
> > **Although the obtained result interesting, It is unclear whether it is of any use for machine learners/computer vision algorithms developers. i think the authors should provide a discussion how their fundamental findings can be potentially used in practice.**
>
> While we acknowledge that it is unlikely that deep learning models will be used directly for the purpose of shape classification, we believe that the *fundamental disparities* we demonstrate in this work between human and machine vision are of critical interest to the community. Given the ever-increasing importance of AI alignment, especially with rapid and unchecked progress in the field, understanding and correcting for misaligned model behavior on our dataset is a worthwhile endeavor for machine learning researchers. We have updated the introduction to reflect this.
>
> > **Irrelevance to the datasets and benchmark track (?) Overall, I do not dot understand why this paper is submitted to the benchmark and datasets track. To be precise, the dataset created here is rather simple (and, in my opinion, this dataset is not a significant contribution) and it seems to me that the main contribution is their empirical findings about neural networks. Hence the current paper may better suit the main conference track or, even better, some other conference or journal, possibly related to human/AI perception etc.**
>
> We appreciate the reviewers’ perspective on the relevance of our work to the Datasets and Benchmarks track. However, based on the relevant topics listed in the official Call for Papers, we view our work as not only contributing 1) a highly scalable, flexible, and efficient data generation process, but critically 2) *“systematic analyses of existing systems on novel datasets yielding important new insight.”*
>
> Furthermore, a dataset that is seemingly simple does not make it worthless as a dataset; in fact, we believe the fact that neural networks exhibit peculiarities on a simple dataset is more interesting than if they had exhibited peculiarities on more complex datasets. If models do not perform as expected even on our dataset, how can the community trust them for scenarios in the wild?
>
> > **Bitterman (1987) seems to be the only reference supporting the claim that humans better rely on corners than edges. This is not super convincing and I recommend to find more citations here.**
>
> While Biederman was the first to study this phenomenon, there was much follow-on work from that era – we have updated the Related Work to include these sources.
>
> > **I am not an expert in this field of neural networks perception and can not carefully access this aspect.**
>
> We thank the reviewer for their honesty in this regard – nonetheless, we appreciate their feedback and have incorporated relevant components of it in our revision!

---

> > ### Comment · Area_Chair_z55C · 2023-08-22
> > **RE: Scope of the Paper**
> >
> > I would briefly comment that I do believe this paper is within scope of the Datasets and Benchmarks track. In particular, simple datasets that expose failure modes of modern methods are crucial tools we can use as a community to improve our algorithms. Complex, large datasets are important for different reasons, but they do not replace these tasks which address specific limitations of current methods. I urge the reviewer to update their review to de-weight their concerns about relevance to the track.

---

> > > ### Comment · Reviewer_d9g1 · 2023-08-26
> > > **Scope of the paper**
> > >
> > > Dear meta-reviewer and authors,
> > >
> > > I find the insights found in the paper interesting and the experiments to be rather sufficient to support them (as I wrote in my initial review). Since the meta-reviewer asks, I have updated my review and de-weighted the "relevance to the track" component accordingly.

---

### Author Response · Authors · 2023-08-27
**Summary of New Revision and Rebuttals**

We thank the reviewers for their thoughtful comments on our submission. In response to these comments, we have uploaded a new revision of our submission. In particular, this updated paper adds a new figure (Figure 4) that includes confusion matrices produced for a finer-grained analysis of model behavior. Figures 3 and 5 are also updated to include error bars across multiple training runs.

Furthermore, we have re-written the Related Work to include additional references to support Biederman’s original observations as well as references regarding Gestalt closure and how it critically differs from RBC and image recoverability. We have also updated the Introduction to emphasize the fundamental importance of the misalignment phenomenon that our dataset introduces, as well as highlight the importance of our dataset as a minimum viable task that induces this human-machine vision misalignment.

---

> ### Author Response · Authors · 2023-08-29
>
> We thank all the reviewers for their comments and feedback so far. Since the response period is soon coming to a close, please let us know if you have any further comments on our revised submission and responses. Thanks!

---

### Decision · Program_Chairs · 2023-09-22

**Decision:**

Accept (Poster)

**Comment:**

A well crafted benchmark that raises an interesting question and addresses an important gap in the ML literature. The authors make the observation that, in contrast to studies of human perception, ML models seem to use edges to classify occluded shapes, while humans tend to use corners. To this end, they provide a dataset specifically designed to expose this behaviour in trained models. This misalignment, while documented thoroughly in the work, remains a mystery. While some reviewers raised concerns that this contribution, while interesting, is not useful for the development of high-performance ML systems, it is an exceptional example of a systematic analysis of existing systems on a novel dataset that yields an important new insight. Many reviewers believed that a more thorough study of this problem in follow-up works will yields important insight into the differences between human and machine perception, and all but one reviewer revised their review to accept after the discussion period (the final reviewer did not reply to any rebuttals or follow-up queries).